# Glass Substrate Dust Removal Using 233 fs Laser-Generated Shockwave

**DOI:** 10.3390/mi12111382

**Published:** 2021-11-10

**Authors:** Myeongjun Kim, Philgong Choi, Jae Heung Jo, Kyunghan Kim

**Affiliations:** 1Department of Photonics and Sensors, Hanman University, Daejeon 34430, Korea; myeongjun@kimm.re.kr; 2Department of Laser & Electron Beam Application, Korea Institute of Machinery & Materials, Daejeon 34103, Korea; pgchoi@kimm.re.kr

**Keywords:** LSC, femtosecond laser, shockwave, particles, cleaning

## Abstract

Eliminating dust is gaining importance as a critical requirement in the display panel manufacturing process. The pixel resolution of display panels is increasing rapidly, which means that even small dust particles on the order of a few micrometers can affect them. Conventional surface cleaning methods such as ultrasonic cleaning (USC), CO_2_ cleaning, and wet cleaning may not be sufficiently efficient, economical, or environment friendly. In this study, a laser shockwave cleaning (LSC) method with a 233 fs pulsed laser was developed, which is different from the laser ablation cleaning method. To minimize thermal damage to the glass substrate, the effect of the number of pulses and the gap distance between the focused laser beam and the glass substrate were studied. The optimum number of pulses and gap distance to prevent damage to the glass substrate was inferred as 500 and 20 μm, respectively. With the optimal pulse number and gap distance, cleaning efficiency was tested at a 95% removal ratio regardless of the density of the particles. The effective cleaning area was measured using the removal ratio map and compared with the theoretical value.

## 1. Introduction

Dust cleaning methods in display panel manufacturing evolved from radio corporation of America (RCA) panel cleaning [1,2]. Physical cleaning methods utilizing the waterjet, ultrasonic, or megasonic cleaning, and brush methods, which involve the application of various physical forces to the glass panel, can detach dust from the panel [3,4,5,6]. Another cleaning method is the chemical cleaning method and it involves the application of a chemical solution mixed with Deionized water to the glass substrate, which melts the dust and surface of the glass substrate together [7]. This chemical method is quite effective but cannot be applied to certain manufacturing processes and is not an eco-friendly process.

Previously, the resolution of the display panel pixel was on the order of 10 μm, but newer organic light-emitting diode (OLED) or micro light-emitting diode (MicroLED) panel pixels are much smaller [8], and conventional physical cleaning methods cannot detach exceedingly small dust particles.

An alternative method for physical cleaning that has recently gained traction is the laser cleaning method. Laser cleaning methods include laser ablation cleaning, in which a laser focusses on the required spot on the glass panel and ablates dust particles [9,10,11,12]; steam laser cleaning (SLC), in which steam or water vapor along with a focused laser beam is utilized [13,14]; and laser shockwave cleaning (LSC) [15,16,17,18,19,20,21,22].

In the LSC mechanism, a focused laser beam causes a rapid increase in the local temperature, resulting in the dielectric breakdown in air. The high-pressure and high-temperature plasma decays rapidly, which induces the generation of shockwaves. A strong shockwave generates complex pressure wave fields, resulting in both a drag and lift effect on the dust particles [23,24,25].

Various types of LSC processes have been investigated. The pulse energy of nanosecond-pulsed lasers [18,22,23,26] is higher than that of the ultrashort-pulsed lasers, but the area subjected to the thermal effect broadens [27]. As a result, the gap distance from the laser focal point to the substrate needs to be maintained, and the shock wave stream may decrease.

The laser pulse energy, number of pulses, and gap distance were studied with Al particles on a silicon substrate. A high pulse energy and number of pulses, and a smaller gap distance were verified to increase the cleaning efficiency [16,17,18,19]. Additionally, with a smaller gap distances, the effective cleaning region broadened [18,22,23].

In the present study, a high-peak-power laser with a pulse width of 233 fs was utilized. The laser was magnified 20 times by the objective lens. The focused beam was irradiated in a direction parallel to the glass substrate. SiO_2_ particles mimicked dust material, and their densities were controlled for uniform initial conditions. The pulse number for which thermal damage to the glass substrate does not occur was investigated to increase the cleaning efficiency.

In previous research studies on the LSC process for micro-/nanoparticle elimination, many experimental works were performed to increase cleaning efficiency and they reached an efficiency of about 90% [23,28]. The gap distance within the laser beam propagation field caused thermal damage and the 95% cleaning efficiency was achieved with a 20 μm gap distance. The effective cleaning region was investigated experimentally and compared with the theoretical results.

## 2. System Set Up and Methods

### 2.1. Experimental Set Up

Figure 1 shows the experimental setup for cleaning the dust particles on the glass substrate. The pulse width of the laser system (Pharos, Light conversion, Vilnius, Lithuania) was 233 fs at a wavelength of 1030 nm, with a maximum power of 10 W. The pulse repetition rate was 50 kHz, and the pulse energy was 134 μJ. The laser beam was delivered by the reflecting mirrors, and, finally, it was focused with 20 times magnification by the objective lens. We measured the diameter of the focused beam to be 15.8 μm using a focused beam profiler (FM100-Focus Monitor 100, Metrolux, Berlin, Germany). The objective lens was moved using the x-y-z linear motors and controlled by PC programming. We prepared gorilla glass with a thickness of 400 μm, and the glass substrate was attached to the rotational stage, which was fine-tuned by a micrometer. The air ejector suctioned the glass substrate onto the rotational stage.

The focused laser beam was irradiated in a direction parallel to the surface of the glass substrate with a small gap between the focused beam and the surface of the glass substrate, as shown in Figure 2, where *W*_0_ is the radius of the focused laser beam, *d* is the gap distance, and *l* is the half length of the glass substrate.

Before the LSC process, we wanted to homogenize the density of the particles using the air-spraying method. Initially, we sprayed the SiO_2_ particles manually on the glass substrate, as shown in Figure 3a, after which the air spray was applied to the particles at 0.05 MPa for 1 min. The remaining particles of SiO_2_ are shown in Figure 3b. Air-spray particle cleaning is not sufficient, but this process may help control the initial density of the particles. We prepared ten samples and measured the density and number of the particles. Two samples, having a particle ratio of more than 0.5%, were eliminated. The average particle ratio was 0.4%, and the standard deviation among the eight samples was 0.05%.

Particle analyzer software (InnerView 2.0, Innerview Co., Ltd., Seongnam, Korea) was used to analyze particle size and density. We analyzed the distribution of particles by size, as shown in Figure 4. Particles less than 3 μm in size occupied over 80% of the area. The sizes of the particles varied from 0.8 to 11.4 μm, and the average particle size was 1.7 μm. The ratio of particles occupying an area from a certain field of view was 0.4%, and the results are summarized in Table 1.

### 2.2. Observation of the Particle Removal Process Using High-Speed Camera

In Figure 5, the particle removal process is visualized using a high-speed Charge coupled device (CCD) camera (Chronos 1.4, Kron Technologies, San Diego, CA, USA). The frame rate of the camera was approximately 10,000 fps and the interval of the frame was 93.65 μs. The pulse-to-pulse interval was 20 ms. In our previous research works for wide area cleaning, thermal damage was not observed with a 70 μm gap during a 2 min laser deposition. The laser beam was focused at the top of the glass substrate with a 70 μm gap. To illustrate the removal process clearly, the quantity of particles introduced was nearly ten times the typical quantities observed in real-life situations. Figure 5a shows the moment before the laser irradiation and removal motions are shown at various time domains in Figure 5, which shows the focused beam shape and movement of the removed particles at different times. The plasma of the focused beam is shown in Figure 5b–h and it reaches the maximum density in Figure 5h. After the force of attachment between the particles and the glass substrate is weakened by the laser shock wave pressure, the particles may fall due to gravity. 
Appendix A shows the animation of Figure 5.

## 3. Results and Discussion

### 3.1. Damage Observation in the LSC Process

Before analyzing thermal damage in the LSC process, the glass substrate and SiO_2_ particles were observed by scanning electron microscope (SEM) images and energy dispersive spectroscopy (EDS) profiles, as shown in Figure 6. The glass substrate with a 300 × 300 μm area was scanned and the element compositions were averaged with the scanning area. The SiO_2_ particle was selected. The difference in the composition of the SiO_2_ particle with that of the glass substrate is that the carbon element appears in the SiO_2_ particle and the percentage of weight of silicon is higher in the SiO_2_ particle.

In the LSC process, the damage to the glass substrate needs to be minimized. Thermal damage with the femtosecond laser pulse is typically limited, but accumulated femtosecond pulses can cause thermal damage. We employed over 20,000 laser pulses with a repetition rate of 50 kHz, and the results are shown in Figure 7. Damage was observed in the focal region, and particles were removed from a donut-shaped region with a radius of approximately 300 μm. Small debris was also observed in the areas far from the focal region of the laser.

The thermally damaged area at one point of the focal region of the glass substrate surface and one point from the debris region was selected, and is shown in Figure 8a,b. The two points were analyzed using EDS profiles, and their compositions were different. The concentration of carbon was quite high in the focal region, as shown in Figure 8a, because it may be caused by the thermal damage induced by the accumulation of pulses. The debris, which was located far from the focal region of the glass substrate, was analyzed using the EDS profiles in Figure 8b. The elements of the debris were quite similar to those of the original glass substrate as shown in Figure 6a. The weight of the elements of both profiles showed similar results. As such, the debris can be assumed to be broken glass from the focal area.

Therefore, it was necessary to minimize the number of pulses to reduce the thermal damage to the glass substrate. Figure 9 shows the results of the cleaning for different pulse numbers. The cleaning effect with 30 laser pulses is not obvious in Figure 9b, but after 110 pulses were employed, almost all the particles inside the rectangular region were removed. Thermal damage was not observed until 110 pulses were applied. As the pulse numbers varied, the particle removal ratio is plotted in Figure 10a. When the applied pulse numbers exceeded 30, the particle removal ratio became approximately 80%, almost reaching saturation. The remaining particles were classified by size after cleaning, as shown in Figure 10b,c. When the number of applied pulses was 30, particles larger than 2 μm were removed, but the number of particles smaller than this increased. This may be because of the shockwave-induced breakage of the larger particles, which in turn may have generated smaller particles. As the number of applied pulses increased to 110, particles larger than 1 μm in size were mostly removed, but particles smaller than 1 μm remained.

### 3.2. Damage Observation in LSC Process

To remove particles smaller than 1 μm in size, the gap distance must be optimized. As shown in Figure 2, the gap distance required to avoid contact with the glass substrate was predicted assuming the theoretical Gaussian beam propagation. The radius of the focused beam, *W*_0_, was measured to be 7.9 μm using a focused beam profiler (FM100-focus Monitor 100, Metrolux, Berlin, Germany). Using this beam diameter, the gap distance required to avoid contact with the surface of the glass substrate (*l* = 300 μm) was predicted to be 14.4 μm. Subsequently, we investigated the damage to the glass substrate at *d* = 10 μm and 20 μm, as shown in Figure 11. Three selective positions on the glass substrate at *d* = 10 μm after 110 pulses were applied are shown in Figure 11a. The top of the glass substrate was broken, and the damaged area propagated above the laser beam path. However, no damage was observed in any of the four positions when the gap distance was 20 μm, as shown in Figure 11b.

The density of particles was set at values of 0.4%, 0.8%, and 6%, as shown in Figure 12. Compared to the case of 0.4% density, the particles were more concentrated in the center in the case of 0.8%, as shown in Figure 12c. In the 6% density case, particle aggregation was observed. Regardless of the density and concentration of the particles, a laser cleaning efficiency of over 95% was achieved with the previously inferred LSC conditions, with a pulse number of 110 and a gap distance of 20 μm. These results show that the LSC process can be applied when the particle density is high and when there is a possibility of particle aggregation.

### 3.3. Effective Area Prediction of LSC Process

The particle removal ratio map is shown in Figure 13. The 30 × 30 μm^2^ grid was divided for mapping. The laser focal region is highlighted in blue. The area of removal ratio of over 95% can be predicted as the 255 and 107 μm rectangular area. The area with 100% particle removal is mapped in green. In addition, except for the green area, the glass substrate is shown in white, and the removal ratio is less than 100%. we The mapping area is important for predicting the effective region that can be influenced by the laser shock wave. The negative values inside the grids imply that the number of particles increase after the LSC process because the nearby particles have moved. In Figure 13, the particles remaining after the LSC process are counted where they may in a position farther from the laser focal point. The particle distribution by sizes of Figure 13 is plotted in Figure 14.

We increased the number of applied pulses to 1000 while maintaining a 20 μm gap distance, as shown in Figure 15. When the number of pulses was 500, the cleaning efficiency was similar to the case of 110 pulses. However, more particles appeared for the 1000 pulse case. As in the previous case when the pulse number was 20,000, accumulated heat resulted in thermal damage to the glass substrate, and broken glass particles were formed. Both the gap distance and the applied pulse number should be selected carefully to avoid damage to the glass substrate.

If the pressure of the laser shockwave induced by a high-intensity focused laser beam is larger than the adhesion force of the particles to the glass substrate, the particles may become detached. The shockwave velocity, pressure, and radius were calculated from Equations (1)–(3).
(1)$$v=\frac{4}{5}\frac{k}{\gamma +1}{\left(\frac{E}{\rho_{0}}\right)}^{1/5}t^{-3/5}$$
(2)$$P=\frac{8}{25}\frac{k^{2}\rho_{0}}{\gamma +1}{\left(\frac{E}{\rho_{0}}\right)}^{2/5}t^{-6/5}$$
(3)$$R=k{\left(\frac{E}{\rho_{0}}\right)}^{1/5}t^{2/5}$$
where *v* is the velocity of the shockwave; k is a constant; γ is the specific heat ratio of ambient air (1.4); *E* is the laser pulse energy; $\rho_{0}$ is the air density at 24 °C, which is 1.18 kg/m^3^; P is the shockwave pressure; and *R* is its radius. The shockwave pressure versus radius of the particle is plotted at the same instant in Figure 16. The adhesion force of particles less than 50 μm is mainly governed by the van der Waals force [29,30].
(4)$$Critical\;removal\;pressure=\frac{F_{v}}{S}=\left(\frac{hr}{8\pi Z^{2}}+\frac{hr_{0}^{2}}{8\pi Z^{3}}\right)/\pi r^{2}=\frac{h}{8\pi^{2}Z^{2}r}+\frac{hr_{0}^{2}}{8\pi^{2}Z^{3}r^{2}}$$
where $F_{v}$ is the van der Waals force; *h* is the Lifshitz–van der Waals constant, which is calculated as 4πA/3; *A* is the Hamaker constant, which is calculated as $6.5\times 10^{-20}\;J$; *Z* is the atomic separation 4 Å; *r* is the radius of the particle; and $r_{0}$ is the radius of the contact area of the particle. The diameter of the particle was assumed to be 1.7 μm, which was the average particle size in the experimental study. Using Equation (4), the distance at which the critical removal pressure and laser shock wave pressure is equal was calculated to be 355 μm, which means that a particle with a 1.7 μm diameter size can travel until this distance in the corresponding LSC process. This prediction is slightly greater than the experimental results. This theoretical approach assumes that a single particle travels, and the interaction between particles is neglected.

## 4. Conclusions

A high-peak-power 233 fs laser was employed for cleaning dust particles smaller than several micrometers from a glass substrate. The detaching motion of the SiO_2_ particles was monitored using a high-speed camera, where the plasma was observed to be intensified by the focused laser beam. When the number of applied pulses was excessive, damage to the glass substrate was observed even though the focused laser beam was not in contact with the glass substrate. Under these experimental conditions, a pulse number of 500 was chosen. Within the beam propagation field, the glass was broken and damage was observed. The gap distance between the field of beam propagation and the glass substrate was maintained at a distance of 20 μm, and it was confirmed that the LSC process did not damage the glass substrate. With the optimal pulse number and gap distance, the cleaning efficiency was tested at a removal ratio of 95% regardless of the density of particles. The effective cleaning area was measured using the removal ratio map and compared with the theoretical value.

In this study, the LSC process with a 233 fs pulsed laser was verified as a non-contact method that does not damage the substrate and can clean more efficiently than the air spray method. Using this method, it is possible to clean dust particles smaller than 1 μm in size in an extremely short time from an area that is ten times wider than the diameter of the focused laser beam.

Future studies can expand the LSC process to the wide area substrate for commercial application. The incident laser angle can be adjusted, and the laser beam can be scanned by the x-y linear motors. Additionally, the effective cleaning area can be a guide for scanning the speed and pitch of the laser beam.

## Figures and Tables

**Figure 1 micromachines-12-01382-f001:**
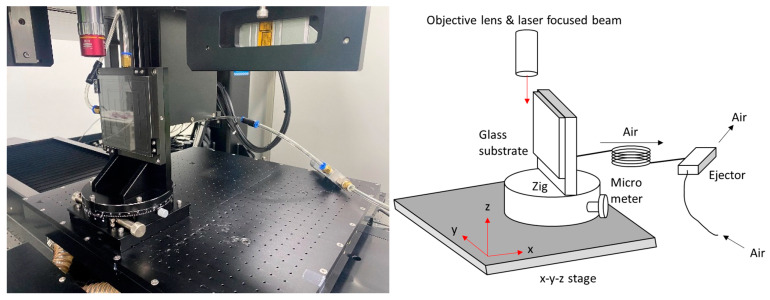
System configuration.

**Figure 2 micromachines-12-01382-f002:**
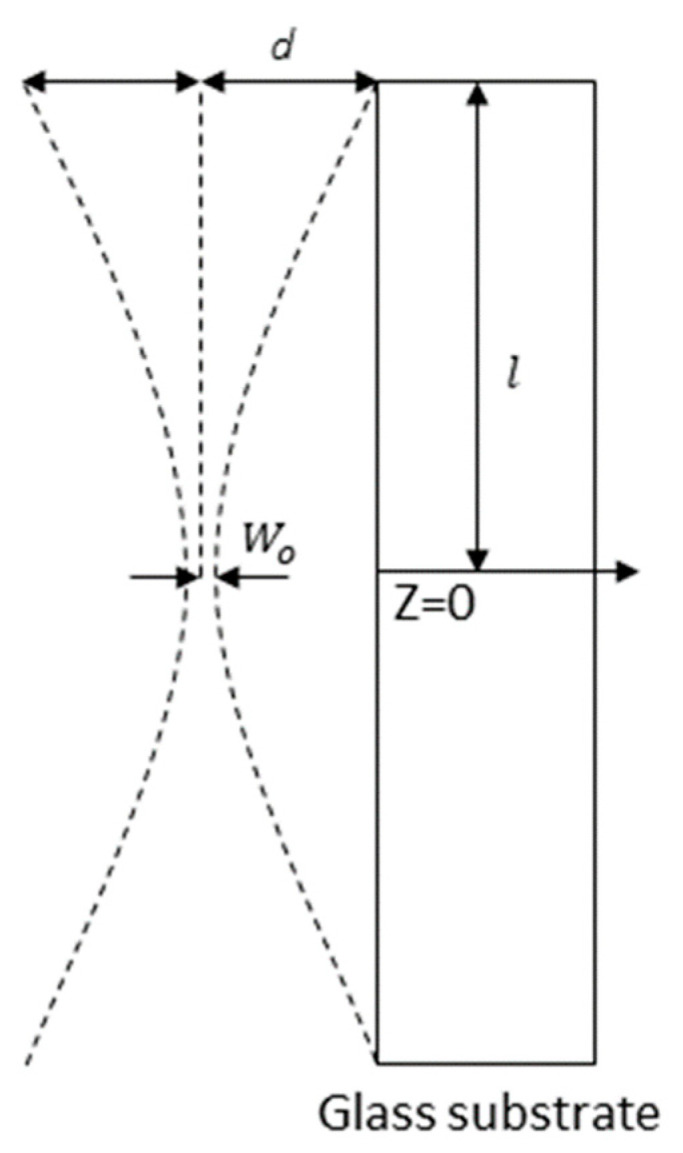
Focused laser beam propagation in the upper glass substrate.

**Figure 3 micromachines-12-01382-f003:**
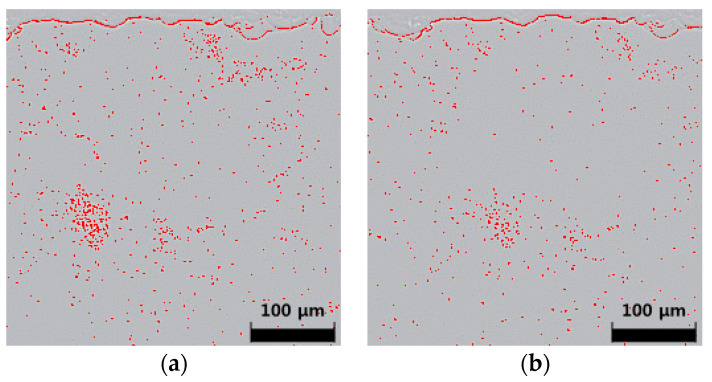
Photos of a glass substrate sprayed with SiO_2_ particles observed using a 10× magnification microscope (MM-800, Nikon, Tokyo, Japan) and particle analyzer software: (**a**) immediately after the application of particles and (**b**) at one minute of air spray cleaning.

**Figure 4 micromachines-12-01382-f004:**
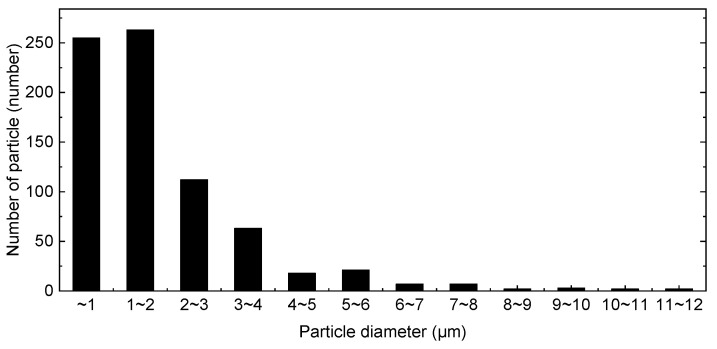
Sample particle distribution based on size.

**Figure 5 micromachines-12-01382-f005:**
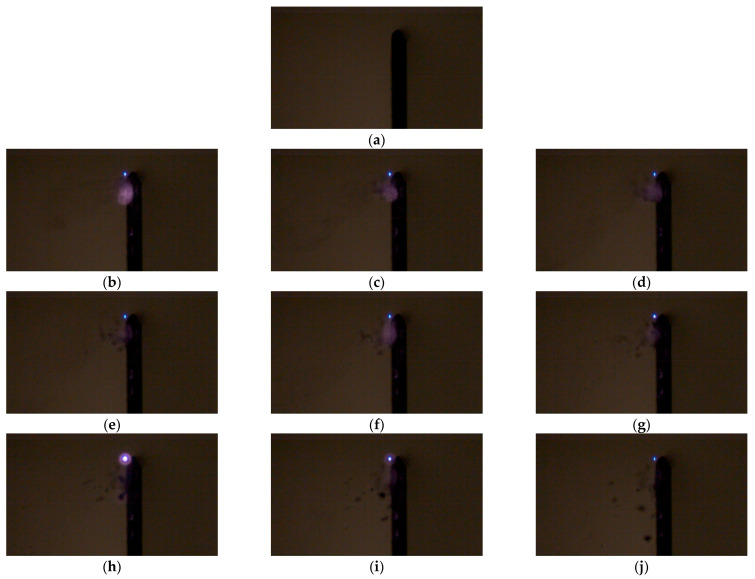
Observation of the particle removal process using a high-speed camera and black bar in glass substrate at (**a**) 0 (**b**) 1.5 (**c**) 3 (**d**) 4.5 (**e**) 6 (**f**) 7.5 (**g**) 9 (**h**) 10.5 (**i**) 12, and (**j**) 13.5 ms.

**Figure 6 micromachines-12-01382-f006:**
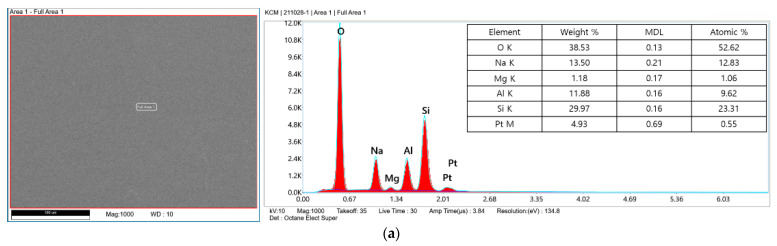
SEM image and particle EDS analysis: (**a**) glass substrate and (**b**) SiO_2_ particle.

**Figure 7 micromachines-12-01382-f007:**
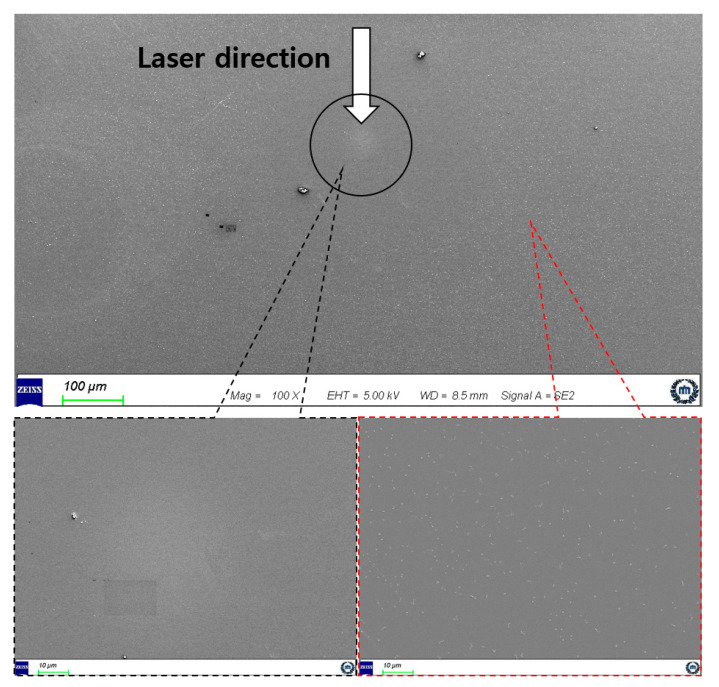
Photo of damage caused by thermal damage and laser ablation: black dotted line area magnification, EHT, WD, signal A is 500×, 5.00 kV, 8.5 mm, SE2, respectively; red dotted line area magnification, EHT, WD, signal A is 1.00k×, 5.00 kV, 8.9 mm, SE2, respectively.

**Figure 8 micromachines-12-01382-f008:**
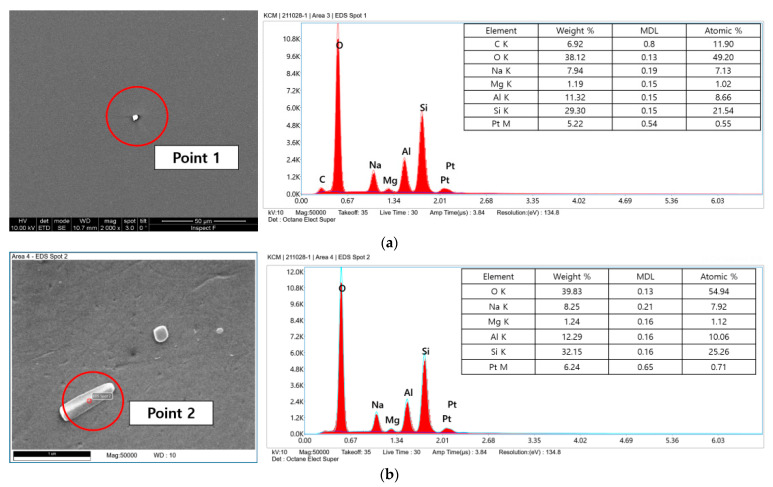
Particle EDS analysis of two points subjected to laser irradiation: (**a**) above focal position and (**b**) assumed broken particles.

**Figure 9 micromachines-12-01382-f009:**
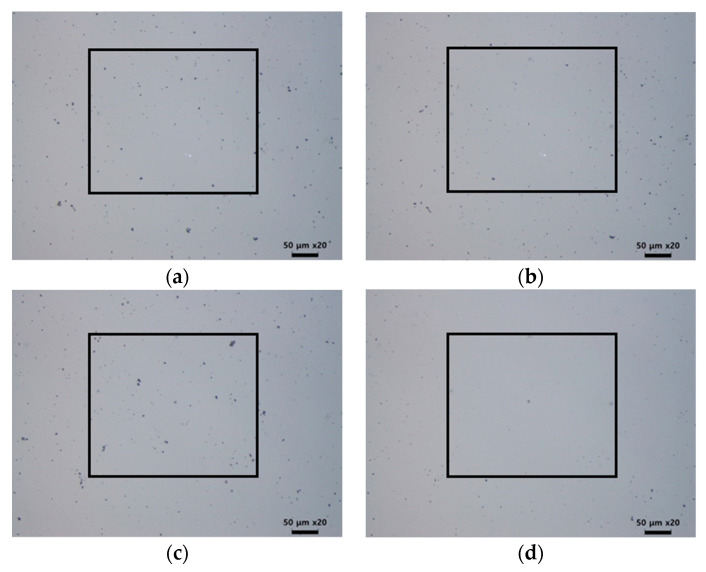
Result of laser cleaning with different pulse numbers: (**a**,**c**) is before cleaning, (**b**) pulse number = 30, and (**d**) pulse number = 110.

**Figure 10 micromachines-12-01382-f010:**
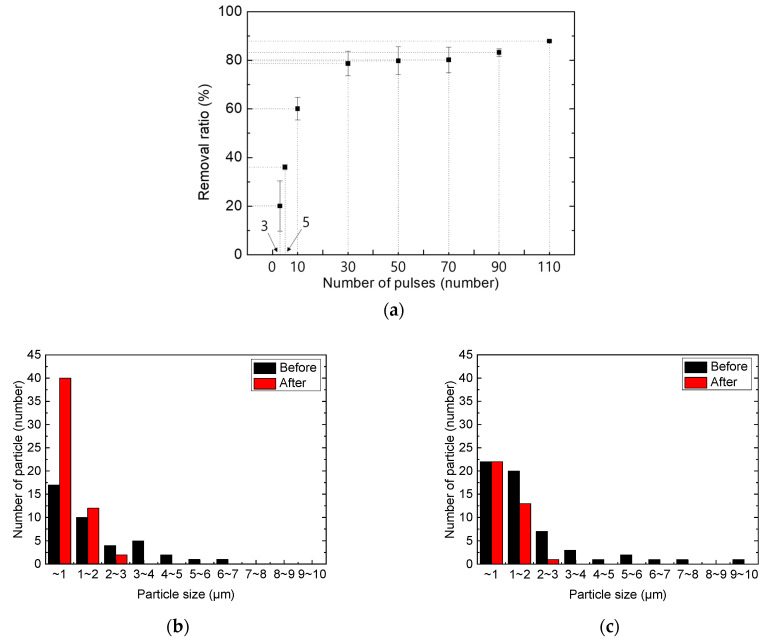
(**a**) Particle removal ratio with number of pulses. (**b**,**c**) Change in particle distribution after LSC process with different pulse numbers: (**b**) pulse number = 30; (**c**) pulse number = 110.

**Figure 11 micromachines-12-01382-f011:**
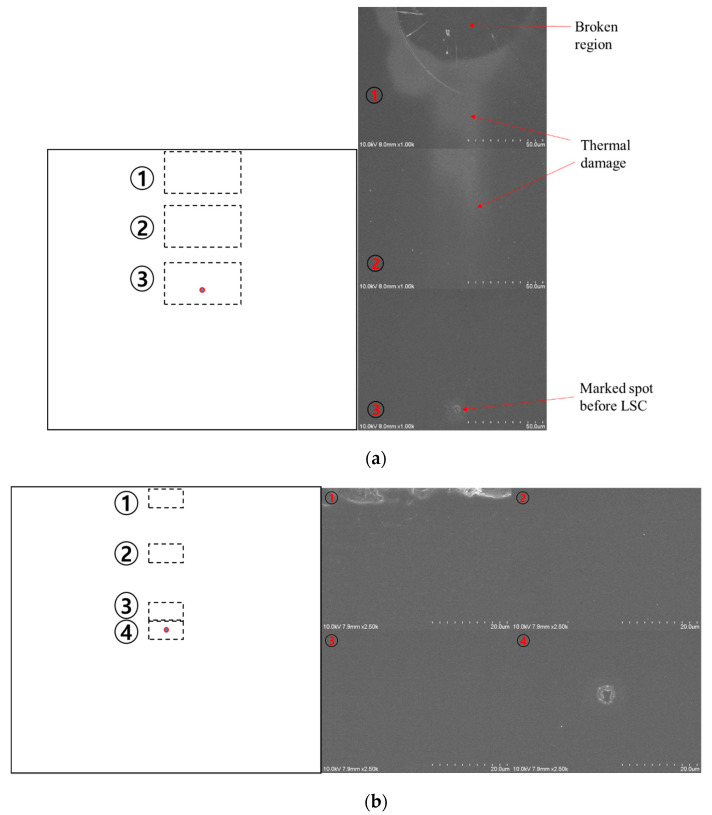
Damage to the glass substrate according to gap distance observed by SEM: (**a**) the gap distance = 10 μm; (**b**) the gap distance = 20 μm.

**Figure 12 micromachines-12-01382-f012:**
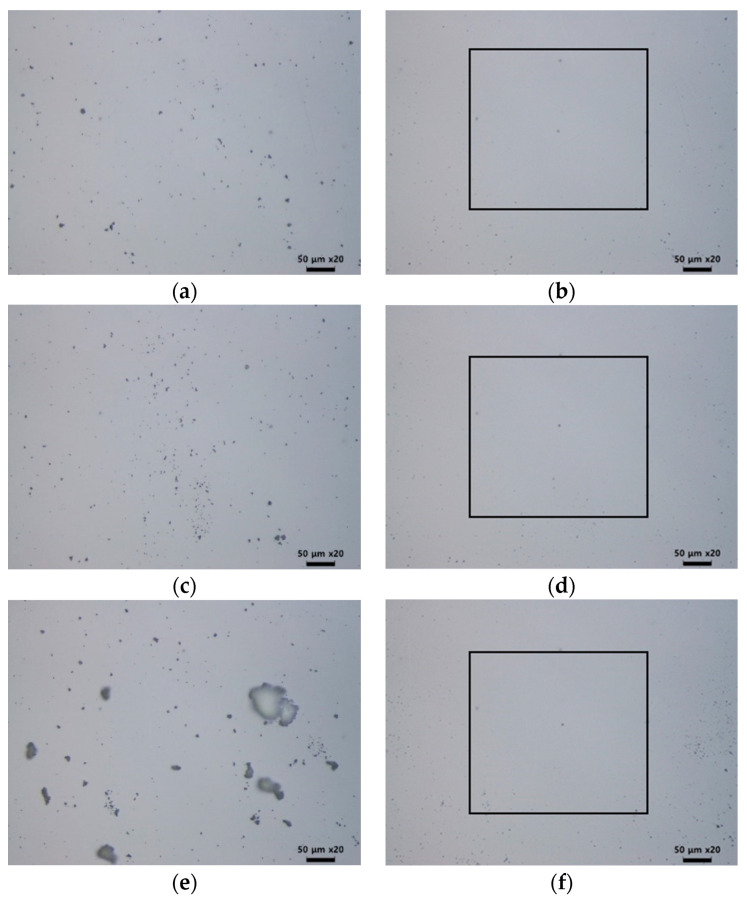
Comparison of laser cleaning efficiency with respect to particle density and distribution (**a**) before cleaning when the particle occupancy rate is 0.4%; (**b**) photo after cleaning of (**a**); (**c**) before cleaning when particle occupancy rate is 0.8%; (**d**) after cleaning of (**c**); (**e**) before cleaning when particle occupancy rate is 6%; (**f**) after cleaning of (**e**).

**Figure 13 micromachines-12-01382-f013:**
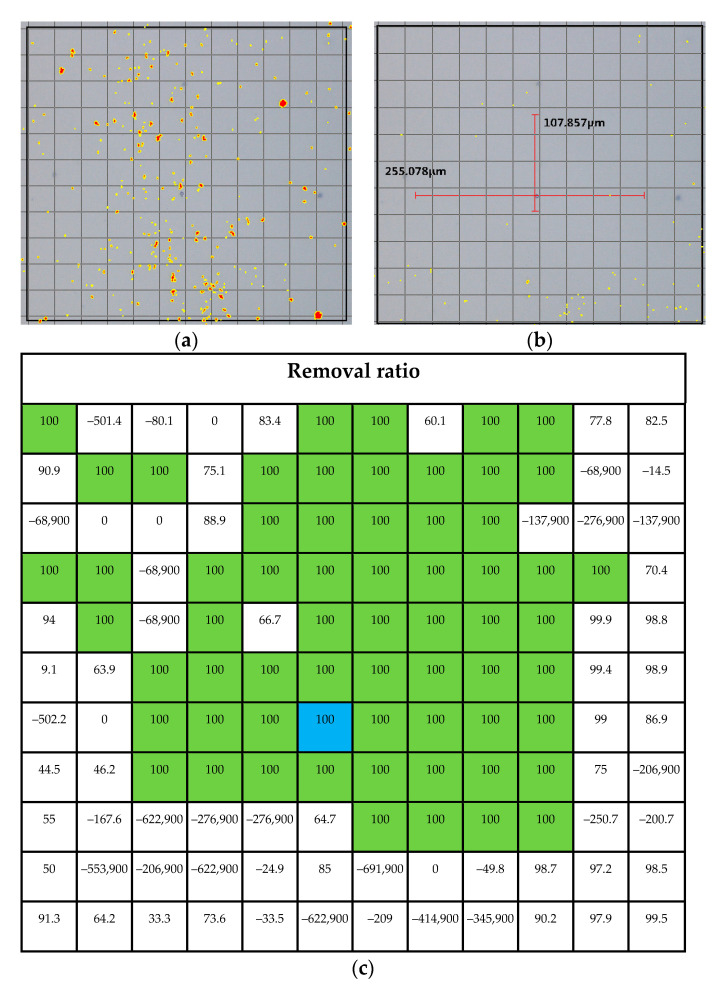
(**a**) Photo of before cleaning when pulse number is 110 number and gap distance is 20 μm, (**b**) photo after cleaning of (**a**), and (**c**) particle removal ratio map.

**Figure 14 micromachines-12-01382-f014:**
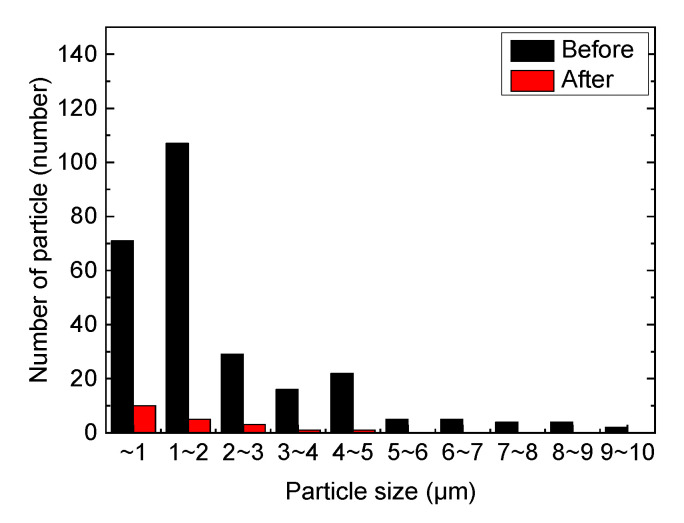
Particle distribution analysis of Figure 13.

**Figure 15 micromachines-12-01382-f015:**
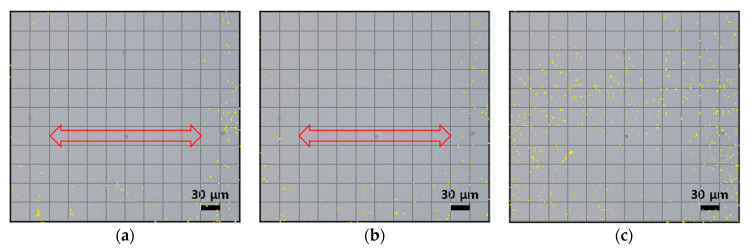
Particle observation caused by laser irradiation when pulse number is (**a**) 110, (**b**) 500, (**c**) and 1000.

**Figure 16 micromachines-12-01382-f016:**
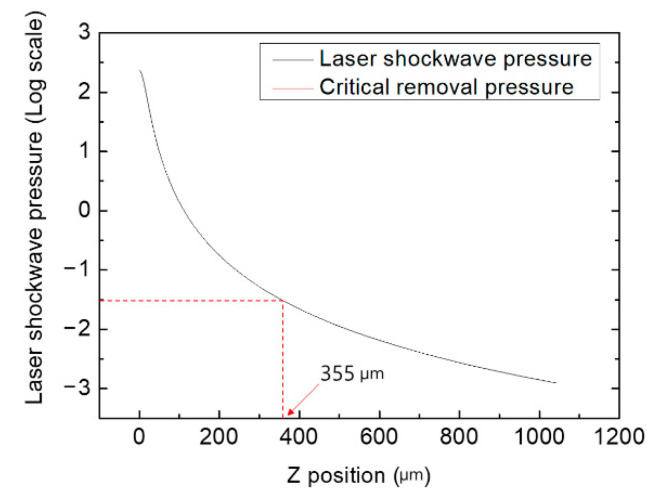
Effective laser cleaning maximum distance by theoretical prediction.

**Table 1 micromachines-12-01382-t001:** Details of the size and density of particles.

List	Particle Parameters
Minimum size (μm)	0.8
Maximum size (μm)	11.4
Average of particle size (μm)	1.7
Analysis area (μm^2^)	1,209,004
Ratio of area occupied by the particles (%)	0.4
Total number of particles (number)	725

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
