# Peer review of "Glass Substrate Dust Removal Using 233 fs Laser-Generated Shockwave"

_micromachines, 2021, doi:10.3390/mi12111382_

Round 1

Reviewer 1 Report

This paper reports the results of laser shock cleaning of particles. The technology itself has a long history and is potentially important for display industry.

However, this manuscript does not present a significant improvement of the technology nor demonstrate a novelty. Most significantly, they used a tightly focused laser beam, as a result, the focus can be brought to the surface only in the region close to the edge (300 μm in the manuscript) but not to the interior area, implying that this technique is difficult to apply for real display devices. The authors must address this issue for consideration of publication.

Besides, there are several parts in the manuscript that are not explained in detail (Figs. 10, 12, & 13 for example). Also, the authors concluded that the particles in Fig. 7 are from the glass substrate. However, it seems that the spectra of SiC particles should also  be shown in comparison in order to make such a conclusion. By theoretical analysis, the authors determined the effective gap between the substrate and laser focus to be 355 μm, which is an order of magnitude difference from the actual gap in experiments (20 μm). It is hard to accept that this difference is slight as the authors claimed. The increase of particles after 1000 pulsed in Fig. 14 and the source of carbon in Fig. 7 are unclear. What is the meaning of total  of particle size in Table 1?

Reviewer 2 Report

The manuscript entitled "Glass substrate dust removal using 233 fs laser-generated shockwave" written by Myeongjun Kim, Philgong Choi, Jaeheung Jo, and Kyunghan Kim provides an interesting alternative to the standard LSC practices.

The manuscript could be considered for publication after clarifying and addressing the following major and minor comments.

Major comments:

1- The authors should clarify why it is preferred the use of a fs pulsed laser over a ps one that is traditionally less expensive and provides larger energies per pulse, all of this without carrying the unwanted thermal effects from ns pulsed lasers.

2- The diagram presented in Figure 2 is pretty inaccurate; it seems that the lens' outgoing beam diameter is in direct contact with the glass surface. Besides, considering the gap distance the authors require to get a proper microparticles detachment, the objective lens should be a bit behind the glass substrate in the X-axis (please refer to the attached figure). Unfortunately, this makes it impossible for the current experimental setup to clean large surface areas, which is ultimately ideal for industrial scalability. In this context, the authors should clarify which are the perspectives for crossing the boundaries between research and application in the real world.

3- Lines 149-151 - It is necessary to take the EDS spectrum of the non-irradiated sample to effectively conclude that the increment of carbon comes from laser-induced thermal damage. Besides, it would be better also to report the at% of each found element to have a proper quantitative comparison.

4- Line 312 - The supplementary material is not mentioned throughout the manuscript, yet it is listed in the Supplementary Materials section.

5- The conclusions should highlight the novelty and importance of the current work for future developments in the field.

Minor comments:

1- Line 40 – References 11 and 12 do not rigorously correspond to LSC procedures.

2- Line 71 – If you used a beam profiler to measure the diameter of the focused beam, please mention it.

3- Line 96 – Please indicate which kind of microscope and parameters were employed to take the images displayed in Figure 3.

4- Line 116 - Why are the authors using a gap of 70 μm when they have determined in section 3.2 that 20 μm is the optimal gap value?

5- Line 121 -  What do the authors want to express with this phrase "The plasma of the focused beam is shown in Fig. 5(h).". The plasma is also observed in images 5 (b)-(j), but 5 (h) is where the plasma seems to reach its maximum density.

6- Figure 10 – Did the damage shown in the top of the glass displayed in Figure 10 (b) was there before the irradiation? If so, why not show the images of the glass substrates before the irradiation?

7- Line 270 – should be in italics, also in equation 3.

8- Line 275 – Please fix the position of the equation number.

Round 2

Reviewer 1 Report

The authors answered most of the questions by the reviewer except the applicability to a large display panel. As an answer to the reviewer's comment to this applicability issue, the authors showed some data in the response letter which are not included in the manuscript though. The explanation in the response letter is not in detail and as a result hard to understand what the authors claim. The large area data should be shown in the manuscript with more details to justify the potential of the developed method.

Author Response

The authors answered most of the questions by the reviewer except the applicability to a large display panel. As an answer to the reviewer's comment to this applicability issue, the authors showed some data in the response letter which are not included in the manuscript though. The explanation in the response letter is not in detail and as a result hard to understand what the authors claim. The large area data should be shown in the manuscript with more details to justify the potential of the developed method.

Answer) I revised by your comments. However, I want to leave the further study for the large area results. Because, the paper is focused the prediction of effective cleaning area and analysis with thermal damage.

Please, be consideration. Thank you.

Reviewer 2 Report

The authors addressed most of the comments. However, keeping in mind that only the highest quality standards are acceptable to publish in a peer-reviewed scientific journal, I consider that the last not properly considered comments should be addressed to recommend the current paper for its publication.

Not appropriately addressed answers:

Major comments:

1- The reviewers should also evaluate the Supplementary Material.

2- Please include the reference to the previous work cited for the 2nd major comment and indicate clearly the primary purpose of the current work with respect to the cited one.

Minor comments (strongly advised to be addressed):

1- Please incorporate the answer provided in response to the 1st major comment.

3- Line 138 – SiO2 --> SiO2

 4- Lines 160-161 – The phrase: “However, the EDS profiles of debris far from the focal region of the laser in Figure 8(b) shows similar those of original glass substrate shown in Figure 6(a).” is not clear, please reformulate it.

5- Please include in the revised version of the manuscript the reasoning behind selecting a 70 μm gap distance for the pictures taken with the high-speed camera.

Author Response

Please, find out attached revised file.

It was revised with blue letters by your comments.

Thank you.
